# Pulsed Four-Wave Mixing at Telecom Wavelengths in Si_3_N_4_ Waveguides Locally Covered by Graphene

**DOI:** 10.3390/nano13030451

**Published:** 2023-01-22

**Authors:** Pierre Demongodin, Houssein El Dirani, Sébastien Kerdilès, Jérémy Lhuillier, Thomas Wood, Corrado Sciancalepore, Christelle Monat

**Affiliations:** 1Université de Lyon, Ecole Centrale de Lyon, INSA Lyon, Université Claude Bernard Lyon 1, CPE Lyon, CNRS, INL, UMR5270, 69130 Ecully, France; 2Université Grenoble-Alpes, CEA-LETI, 17 Avenue des Martyrs, 38054 Grenoble, France

**Keywords:** nonlinear optics, graphene, waveguides, silicon nitride, four-wave mixing, telecom wavelength

## Abstract

Recently, the nonlinear optical response of graphene has been widely investigated, as has the integration of this 2D material onto dielectric waveguides so as to enhance the various nonlinear phenomena that underpin all-optical signal processing applications at telecom wavelengths. However, a great disparity continues to exist from these experimental reports, depending on the used conditions or the hybrid devices under test. Most importantly, hybrid graphene-based waveguides were tested under relatively low powers, and/or combined with waveguide materials that already exhibited a nonnegligible nonlinear contribution, thereby limiting the practical use of graphene for nonlinear applications. Here, we experimentally investigate the nonlinear response of Si${}_{3}$N${}_{4}$ waveguides that are locally covered by submillimeter-long graphene patches by means of pulsed degenerate four-wave mixing at telecom wavelength under 7 W peak powers. Our measurements and comparison with simulations allow us to estimate a local change of the nonlinearity sign as well as a moderate increase of the nonlinear waveguide parameter (*γ*∼−10 m^−1^W^−1^) provided by graphene. Our analysis also clarifies the tradeoff associated with the loss penalty and nonlinear benefit afforded by graphene patches integrated onto passive photonic circuits, thereby providing some guidelines for the design of hybrid integrated nonlinear devices, coated with graphene, or, more generally, any other 2D material.

## 1. Introduction

Graphene is the first and most mature of bidimensional materials that has been isolated [1,2], and it has attracted a lot of attention from the scientific community due to its unique physical and optoelectronic properties [3]. In particular, in addition to its saturable absorption [4] and tunable electro-absorption [5], several studies presented very promising results concerning its Kerr nonlinear optical response at telecom wavelengths (i.e., around 1.5 μm) both theoretically [6,7,8,9] and experimentally [10,11,12]. These findings potentially make graphene a good candidate to improve the performance of nonlinear photonic devices for datacom applications. The nonlinear efficiency of a standard dielectric waveguide is typically quantified by the nonlinear parameter $\gamma =\frac{n_{2}\omega_{0}}{cA_{eff}}$, which roughly gives the nonlinear effect induced per unit of waveguide length and power. By using various types of hybrid graphene/dielectric waveguides, very high effective values of γ have been measured in the literature, by means of self-phase modulation (SPM) [12,13] or four-wave mixing (FWM) [10,14,15,16] experiments. Typical values for γ that range from a few hundred up to a few thousand per watts per meter have been reported around λ=1.5 μm, which is more than one order of magnitude the value achieved with tightly confining silicon waveguides (for instance, [17]).

However, a great disparity exists in the literature regarding the nonlinear response of graphene [18], and the abovementioned results have not led to nonlinear graphene-based hybrid devices with outstanding performance so far [19]. This is partly due to the typically high linear absorption of graphene that tends to mitigate the net nonlinear response of the device, especially at telecom wavelengths. In addition, the origin of the nonlinear effects in graphene, which are mediated by photogenerated carriers at near-IR wavelengths, as was convincingly highlighted in recent papers [20,21,22], seems to restrict the use of the graphene-effective nonlinearity to relatively low power levels, thereby restricting the absolute magnitude of these effects. Practically, if one is to use graphene (or any other 2D material) for locally enhancing the nonlinear response of an otherwise passive photonic integrated circuit, one has to better assess and understand the tradeoff associated with the loss penalty and nonlinear contribution from the resulting hybrid graphene/dielectric section integrated within such circuits.

Although Si is a mature photonic platform, which has been used with graphene in initial demonstrations [11,12,13], it suffers, at telecom wavelengths, from relatively high two-photon absorption, and an associated free carrier penalty which severely limits the resulting nonlinear device performance under increasing powers. Si${}_{3}$N${}_{4}$ on insulator represents another mature platform for creating low loss waveguides with a lower nonlinear response and no two-photon absorption, thereby making it particularly attractive for use with 2D material patches that can locally enhance the relatively modest nonlinear response of the Si${}_{3}$N${}_{4}$ waveguides (*γ*∼1 m^−1^W^−1^).

In this paper, we investigate the use of short graphene/Si${}_{3}$N${}_{4}$ hybrid waveguide sections, and aim to assess the potential of this approach as a way to control and locally enhance the nonlinear response of a mature and low loss Si${}_{3}$N${}_{4}$ waveguide circuit. More specifically, we conduct degenerate four-wave mixing measurements at λ∼1.5 μm on Si${}_{3}$N${}_{4}$ waveguides partially covered with millimeter-scale-long graphene patches to probe the nonlinear response of these waveguides at telecom wavelengths. The specific use of pulsed pump and probe signals with 7 W peak power levels allows us to boost, a priori, the nonlinear effects with respect to prior CW four-wave mixing measurements that have been performed on waveguides covered by graphene under tens of milliwatt powers [10,16]. Our four-wave mixing measurements, using different graphene patch lengths and different pump powers, are compared with simulations that show that graphene locally changes the sign of the nonlinear γ parameter of the waveguide, and enhances its magnitude from 1 m^−1^W^−1^ to −10 m^−1^W^−1^. Our simulations allow us to take into account the global response of the waveguide as well as to differentiate between the linear loss penalty and the nonlinear benefit induced by graphene on the whole waveguide structure. Surprisingly, the outstanding nonlinear response ($|n_{2,gr}|∼$ 10^−13^ m^2^W^−1^ and up to $|n_{2,gr}|∼$ 10^−12^ m^2^W^−1^) of graphene that was reported in a few papers [12,16,23] does not translate, in our measurements, into a tremendously high local nonlinear response of our hybrid graphene/Si${}_{3}$N${}_{4}$ waveguide. The modest nonlinear enhancement afforded by our hybrid graphene/Si${}_{3}$N${}_{4}$ waveguides does not fully compensate for linear loss penalty induced by graphene absorption along the hybrid waveguide patch. Most critically, our work contributes to assessing the nonlinear performance and limits of waveguides locally covered with graphene patches within passive photonic circuits at telecom wavelengths, and their potential in applications.

## 2. Description of the Experimental Conditions

### 2.1. Hybrid Graphene/Si${}_{3}$N${}_{4}$ Waveguide Fabrication and Linear Properties

For this study, we use Si${}_{3}$N${}_{4}$ waveguides ( 1.5 μm wide and 800 nm high) clad with a 2.2 μm thick silica layer. First, deposition of a Si${}_{3}$N${}_{4}$ film was achieved via low-pressure chemical vapor deposition (LPCVD) by using the twist and grow approach [24] for strain management and crack prevention. Next, deep ultraviolet lithography and fluorine-based dry etching were employed for patterning the low-loss Si${}_{3}$N${}_{4}$ waveguides. High-temperature annealing in oxygen and nitrogen atmosphere are applied post-etching to reduce the Si${}_{3}$N${}_{4}$ absorption [25]. A silica upper cladding was then deposited by using high-density, plasma-enhanced chemical vapor deposition (HDP-PECVD), followed by opening a window in it down to the top surface of the Si${}_{3}$N${}_{4}$ waveguides via deep ultraviolet lithography and dry etching process.

Following this fabrication process, the top cladding of the waveguides was thus removed selectively (see Figure 1a,b) in order to expose the core of the Si${}_{3}$N${}_{4}$ waveguides across a specific area along the length of the waveguide. Commercial graphene grown by chemical vapor deposition (CVD) was transferred by Graphenea https://www.graphenea.com/ onto the chip containing the waveguides with the patterned upper cladding. Although graphene covers the whole chip, this patterning allows us to restrict the interaction between graphene and the guided mode of the waveguides solely along the etched windows. This method ensures a relatively good control of both the position and length of the graphene area interacting with the waveguides, without additional post-processing steps after the graphene transfer, which might otherwise affect the graphene optical properties.

The waveguides are 2 cm long in total, and the length of the etched windows, which is located 2 mm far from the edge of the waveguide, varies between 0.2 mm, 0.5 mm, 0.8 mm, 1.1 mm, and 1.4 mm (Figure 1b). A TE-polarized CW signal at λ=1547 nm (around 6 μW coupled) is butt-coupled from the left of the chip (Figure 1a), and first propagates along 2 mm of SiO${}_{2}$-clad Si${}_{3}$N${}_{4}$ before reaching the graphene-covered area. The coupling loss per facet of the chip is estimated to be ∼3 dB, according to our reference measurements on a fully clad Si${}_{3}$N${}_{4}$ waveguide. Performing transmission measurements of the waveguides covered by the different graphene lengths allowed us to extract a value for the linear losses induced by graphene of 86 dB cm^−1^ (see Figure 1c). Under our experimental conditions, no power-dependent transmission was experimentally detected, and the loss thus remained constant for the whole range of power investigated, thereby ruling out any significant saturable absorption of graphene. This remains consistent with our previous work [4], in which pulse energies of approximately 50 pJ were needed to obtain a significant (>10%) variation of graphene absorption. In the present work, the pump pulse energy is at most of 14 pJ, which is most likely not enough to produce a significant variation of graphene absorption. Considering that the propagation loss of the fully clad Si${}_{3}$N${}_{4}$ waveguides without graphene is 0.5 dBcm^−1^, the large loss in the presence of graphene confirms its interaction to be significant with the waveguide mode along the etched windows. One can notice that under those conditions, the effective length (defined as $L_{eff}=\left(1-e^{-\alpha L}\right)/\alpha$) along which nonlinear effects can accumulate in the hybrid graphene/Si${}_{3}$N${}_{4}$ section varies from $L_{eff}=0.16\;mm$ (for $L_{gr}=0.2\;mm$) up to $L_{eff}=0.45\;mm$ (for $L_{gr}=1.1\;mm$).

Moreover, in order to ensure the quality of the graphene transferred onto the chip, Raman measurements were conducted after the transfer, and confirmed a high-quality monolayer graphene [26], with the absence of the typical D peak (at 1350 cm^−1^). We estimate a Fermi level of approximately −0.3 eV corresponding to an approximate p-doping of 9 × 10^12^ cm^−1^, expected for CVD graphene and this transfer process [27].

### 2.2. Four-Wave Mixing Experiments

The pump-probe experiments are performed by using a pulsed laser at telecom wavelengths that delivers 2 ps pulses centered around 1547 nm with a repetition rate of 20 MHz. From this input signal, Figure 2 shows the different steps that allow the generation of synchronized TE-polarized pump and probe pulsed signals that are slightly detuned in wavelength and whose power can be changed independently. The TE polarization is maintained along the different optical fibers. The setup consists of an amplifier and a programmable spectral filter that can control the spectral bandwidth and detuning of the pump and probe signal, created from the spectrally broadened pulsed input laser signal. The spectral detuning between the pump and probe is set at 4 nm and the bandwidth of each signal is 2 nm with a relatively sharp frequency cut.

The interaction of the pump ($\lambda_{p}=1545\;\mathrm{nm}$) and the probe ($\lambda_{s}=1549\;\mathrm{nm}$) will generate an idler signal of approximately $\lambda_{i}=2\times \lambda_{p}-\lambda_{s}=1541\;nm$ by degenerate four-wave mixing along the waveguides. The probe power is fixed in our experiments (coupled peak power of 7 W), whereas the coupled peak power of the pump is varied between 1.3 W and 7W.

## 3. Nonlinear Measurements on the Hybrid Waveguides

In order to characterize the nonlinear response of the graphene-covered section of the waveguide, we probe the idler generation as a function of both the peak pump power and the length of graphene covering the waveguide. Figure 3 shows the spectrum measured for the waveguides covered by different graphene lengths, and for the case where the coupled peak power $P_{pump}=P_{probe}=7W$.

The idler signal is clearly detected at approximately 1541 nm. As the graphene length in contact with the Si${}_{3}$N${}_{4}$ waveguide increases, we observe an overall reduction of the whole signal (probe, idler, and pump). This signature is directly correlated to the high linear losses induced by the graphene-covered section of the waveguide as was measured in the Section 2.1. The four-wave mixing conversion efficiency (*CE*’), after propagation along a waveguide of length *L*, is usually defined as [28]
(1)$$CE^{\prime }=\frac{P_{idler}\left(L\right)}{P_{probe}\left(0\right)},$$
where the input probe power at the entrance of the waveguide is typically considered. In our case, each waveguide has a different drop in transmission, depending on the graphene length covering it (between ∼2 dB for $L_{gr}=0.2\;mm$ and ∼9 dB for $L_{gr}=1.1\;mm$), which is relatively constant with wavelength across the C-band and therefore equally affects the pump, probe, and idler signals. Therefore, to somewhat separate this linear loss penalty from the FWM conversion efficiency and quantitatively analyse the impact of the nonlinear response of graphene on the idler generation, we use instead the following ratio, referred to as the FWM conversion efficiency in the rest of the paper:(2)$$CE=\frac{P_{idler}\left(L\right)}{P_{probe}\left(L\right)}.$$

This expression gives higher values than Equation (Equation 1) for the *CE* because the probe power at the output of the waveguides is decreased with respect to that at the entrance of the waveguides by the propagation loss. Equation (Equation 2) thus allows us to leave aside the graphene-induced optical power drop equally affecting the output signals from the *CE* estimation. Note that this ratio can also be directly extracted from the measured FWM spectra.

Figure 4 shows the conversion efficiency given by Equation (Equation 2) as a function of the graphene length interacting with the waveguides, and for different coupled pump powers. We first observe that the *CE* increases with the coupled pump power for each waveguide (Figure 4a), as expected, and that this increase can be well fitted with a quadratic behavior (dotted line on Figure 4a). Regarding the impact of graphene, we observe on Figure 4b that for each pump power, the *CE* decreases with the greater length of graphene. Furthermore, the maximum drop of *CE*, measured by comparing $L_{gr}=1.1\;mm$ with the reference waveguide, increases in amplitude with power and varies from −11 dB down to −15 dB, for 1.4 W and 7 W pump power, respectively.

At first sight, various phenomena might explain the unexpected and apparent *CE* reduction caused by graphene in these experiments. The first and simpler explanation is the high linear propagation loss caused by graphene, which might hide a potential increase of the nonlinear Kerr response along the hybrid graphene/Si${}_{3}$N${}_{4}$ section compared to the bare Si${}_{3}$N${}_{4}$. Another explanation might be found in the sign of the nonlinear contribution of graphene, which was measured to be negative [12,23], i.e., of opposite sign to the Si${}_{3}$N${}_{4}$ waveguide nonlinear response before and after the hybrid graphene/Si${}_{3}$N${}_{4}$ section. The fact that the whole chip consists of three subsequent and distinct waveguide sections, respectively without/with/without graphene, indeed makes it more difficult to directly account for both the linear and nonlinear local contribution of graphene to the cumulative four-wave mixing response measured across the entire chip. Therefore, to tell these different effects apart, we carry out some simulations in the next section, which take into account the response of the hybrid graphene/Si${}_{3}$N${}_{4}$ waveguides, and that of the bare Si${}_{3}$N${}_{4}$ waveguides before and after the graphene-covered waveguide section. These simulations will thus clarify the impact of the linear loss and nonlinear response of graphene as induced locally within the hybrid graphene/Si${}_{3}$N${}_{4}$ waveguide section on the response of the whole structure.

## 4. Comparison with Simulations and Discussion

Our waveguides are composed of three consecutive sections of waveguides (clad Si${}_{3}$N${}_{4}$; unclad graphene covered Si${}_{3}$N${}_{4}$; and clad Si${}_{3}$N${}_{4}$). We model the degenerate four-wave mixing response of each waveguide section by a coupled system of nonlinear Schrödinger equations involving the interacting pump, probe, and idler signals. The system of Equation (Equation 3) describes the evolution of the electric field envelope, $A_{j}$, with $j\in \left\{p,s,id\right\}$ for the pump, probe, and idler, respectively [28]. We have
(3)$$\left\{\begin{array}{cc} \; & \partial_{z}A_{p}+i\frac{\beta_{2}}{2}\partial_{t}^{2}A_{p}=-\frac{\alpha_{p}}{2}A_{p}+i\gamma |A_{p}{|}^{2}A_{p}+2i\gamma {\left|A_{s}\right|}^{2}A_{p}\\ \; & \partial_{z}A_{s}+\delta_{s}\partial_{t}A_{s}+i\frac{\beta_{2}}{2}\partial_{t}^{2}A_{s}=-\frac{\alpha_{s}}{2}A_{s}+i\gamma |A_{s}{|}^{2}A_{s}+2i\gamma {\left|A_{p}\right|}^{2}A_{s}\\ \; & \partial_{z}A_{id}+\delta_{i}\partial_{t}A_{id}+i\frac{\beta_{2}}{2}\partial_{t}^{2}A_{id}=-\frac{\alpha_{i}}{2}A_{id}+2i\gamma |A_{s}{|}^{2}A_{id}+2i\gamma {\left|A_{p}\right|}^{2}A_{id}+i\gamma A_{p}^{2}A_{s}^{☆}. \end{array}\right.$$

$\partial_{t}$ and $\partial_{z}$ represent the temporal and spatial derivative, respectively. *i* is the imaginary unit, $\beta_{2}$ represents the dispersion coefficient of the second order, $\alpha_{j}$ the linear propagation loss (kept constant here for all three signals), and γ the nonlinear parameter of the considered waveguide section. $\delta_{j}=\beta_{2}\Delta \omega$ is associated with the probe or idler walk-off, in the frame of the pump. Graphene saturable absorption was not included in Equation (Equation 3), because we did not observe any signature of this effect under our experimental conditions.

We use the split-step Fourier method (SSFM) to numerically solve this system of equations. It allows us to change the parameters along the propagation direction according to the specific response of the local waveguide structure, while feeding the output signal of one given section as the input for the simulation of the subsequent one. Table 1 contains the parameters used in the simulations to model each waveguide section (with and without graphene). The second-order dispersion, effective area, and nonlinear parameter were computed by using mode profile simulation with the software Lumerical^®^.

The loss at the etched interface corresponds to the estimated loss when the guided mode crosses the boundary between the clad section of the Si${}_{3}$N${}_{4}$ waveguide and the unclad one (i.e., covered with graphene). By using these parameters, we run simulations in order to identify which value for the nonlinear coefficient $\gamma_{hybrid}$, associated with the hybrid graphene/Si${}_{3}$N${}_{4}$ section of the waveguide, best reproduces the four-wave mixing measurements for all graphene lengths and pump power values. The simulation results are shown in Figure 5 along with the measured *CE* as a function of the graphene patch length, and for different coupled pump powers.

From the simulated curves, the case $\gamma_{hybrid}=\gamma_{Si_{3}N_{4}}$ (yellow curves) is equivalent to considering that graphene has no particular nonlinear contribution to the whole waveguide response, but only affects it negatively via adding some linear loss along the hybrid graphene/Si${}_{3}$N${}_{4}$ section. We obtain from these curves the direct loss penalty caused by graphene on the four-wave mixing response of the whole waveguide. Accordingly, graphene-induced losses give rise to a drop in the conversion efficiency of approximately 10 dB between the 2 cm-long Si${}_{3}$N${}_{4}$ structure without graphene and the Si${}_{3}$N${}_{4}$ waveguide locally covered by 1.1 mm of graphene. This *CE* decrease primarily reflects the missing nonlinear contribution from the 1.7 cm-long Si${}_{3}$N${}_{4}$ waveguide following the hybrid graphene/Si${}_{3}$N${}_{4}$ section, whenever the latter strongly absorbs the signal, i.e., for increasing graphene lengths. However, the measurements (in black) suggest an even stronger reduction of the conversion efficiency induced by the presence of graphene, which reaches up to −5 dB with respect to the yellow curve, for the maximum pump power (7 W) and graphene length (1.1 mm).

As observed on Figure 5, the four-wave mixing *CE* measurements are relatively well reproduced considering −10 m^−1^W^−1^ < $\gamma_{hybrid}$ < −7 m^−1^W^−1^ i.e., an absolute value almost one order of magnitude larger than that of the clad Si${}_{3}$N${}_{4}$ waveguide. This single value for $\gamma_{hybrid}$ consistently reproduces the measurements for the whole range of graphene lengths and pump powers shown in Figure 5, apart from the sole 1.4 W measurement, which is a bit less reliable due to lower S/N and the lack of sensitivity of our setup. This estimated $\gamma_{hybrid}$ value is also found to be negative, because positive values would further deviate from the measured trend as compared with the case $\gamma_{hybrid}$ = $\gamma_{Si_{3}N_{4}}$ (yellow curve). Considering that the nonlinear response of Si${}_{3}$N${}_{4}$ yields a positive nonlinear parameter, a negative effective $\gamma_{hybrid}$ implies that the nonlinearity induced by graphene itself is negative, as was suggested by earlier reports [12,23], and is strong enough to overcompensate the nonlinearity of the underlying Si${}_{3}$N${}_{4}$ waveguide on which it is deposited. Qualitatively, this *CE*-enhanced reduction can be understood by the opposite contributions to the idler signal generation arising, respectively, from the hybrid graphene/Si${}_{3}$N${}_{4}$ section (with a negative $\gamma_{hybrid}$) and the clad Si${}_{3}$N${}_{4}$ sections (with a positive $\gamma_{Si_{3}N_{4}}$) before and after the section covered by graphene. Although our measurements and simulations allow us to quantify and demonstrate some nonlinear enhancement of the waveguide locally provided by graphene, this suggests that our particular Si${}_{3}$N${}_{4}$ chip is here not ideal to practically exploit the nonlinear effects of graphene, as the contributions of consecutive sections with opposite signs are undoing each other.

A comparison of our results with the literature on graphene-covered nonlinear waveguides is shown in Table 2. The references used here correspond to similar SPM and FWM experiments at telecom wavelengths, yet under slightly different experimental conditions (either CW or pulsed signals and various power levels), as indicated in the table. Regarding first the sign of the nonlinear response of graphene that was extracted from the different experiments, our negative value is consistent with the reports from Vermeulen’s group [12], but it is the opposite of what was found by others. We argue, in particular, that the extraction of graphene nonlinearity is quite tricky from measurements on hybrid graphene/Si waveguides, in which the positive nonlinear contribution from the underlying Si waveguides (almost two orders of magnitude larger than for Si${}_{3}$N${}_{4}$ waveguides) cannot be ignored. We also highlight that it is not straightforward to extract the sign of the nonlinearity from FWM measurements. In our case, the opposite nonlinear contribution from the Si${}_{3}$N${}_{4}$ waveguide sections (for which $\gamma_{Si_{3}N_{4}}>0$) combined with our simulations enabled us to reliably access the negative value for the effective nonlinear parameter of our hybrid graphene/Si${}_{3}$N${}_{4}$ section.

Focusing on the four-wave mixing measurements presented in the Table 2, our work is conducted in a pulsed regime rather than by using CW signals, allowing us to test the waveguides under several watts instead of tens [10] or hundreds of milliwatt [16]) powers, which should boost, in principle, the nonlinear effects observed. According to the usual perturbative description of the nonlinear response of dielectric materials, the power variation should yet not affect the hybrid waveguide nonlinear response. We observe, however, a striking difference between the effective nonlinear $\gamma_{hybrid}$ for the graphene-covered waveguides, which is, in our case, of approximately 10 m^−1^W^−1^, i.e., several orders of magnitude less than the other values indicated in the Table, where up to a few thousand were reported. Considering the 2D nature of graphene covering the waveguide and assuming that the nonlinear response of graphene is a few orders of magnitude greater than the one of the underlying waveguide (which is particularly true for Si${}_{3}$N${}_{4}$), one could rightly argue that the net nonlinear effect of the hybrid section is constrained by the interaction between the guided mode and the graphene covering the waveguide. By using the effective thickness approach to describe graphene as a thin (typically ∼0.3-nm thick) but standard material, an equivalent nonlinear effective index $n_{2,graphene}^{eff}$ can be inferred from the mode field distribution and its overlap with graphene [11]. Although this approach is questionable for 2D materials, and a more relevant approach could be considered [22], it allows us to more simply compare the values inferred for the graphene nonlinear response between different hybrid waveguides, after factoring out the variations in the light–graphene interaction between the different underlying waveguide geometries used in Table 2. By using this approximated approach, our results suggest an equivalent nonlinear index for graphene of $n_{2,graphene}^{eff}∼-10^{-14}$ m^2^W^−1^. This value remains one order of magnitude lower than the one extracted from the references in Table 2, which lead to $|n_{2,graphene}^{eff}|∼10^{-13}$ m^2^W^−1^ for most of them.

From a practical point of view, to evaluate the efficiency of graphene-covered waveguides within photonic circuits, we should take into account both the additional loss penalty and the nonlinear contribution provided by the local addition of graphene. The graphene-induced linear propagation loss of the hybrid mode (denoted as $\alpha_{hybrid}$) reasonably reflects the degree of interaction between the guided mode and graphene. This loss varies quite significantly between the different waveguide geometries of Table 2, and ranges between 86 dB cm^−1^ for our work up to several hundred and over 1000 dB cm^−1^ for others. To some extent, the nonlinear contribution of graphene to the overall nonlinear response of the hybrid guided mode should similarly increase with this interaction. It is thus relevant to compare the ratio $\gamma_{hybrid}$/$\alpha_{hybrid}$ for the different reports so as to quantify the amount of nonlinear effects that a particular hybrid graphene/dielectric structure can produce per unit of power, normalised with respect to the associated graphene-induced loss penalty. The last column of Table 2 shows this quantity for the different graphene/dielectric waveguide geometries. This ratio remains much weaker, in our case—between one and two orders of magnitude lower with respect to the other references. Therefore, despite the relative lower loss of our structures, reflecting a weaker interaction of graphene with the guided mode, much lower nonlinear effects can be comparatively achieved. In our geometry, the gain in the nonlinear parameter thus remains marginal relative to the loss penalty locally induced by graphene.

Having thus factored out the effect of the loss and the light–graphene interaction, we are left to explain the striking difference in the graphene nonlinear response from that of the literature. Although the perturbative approach for describing nonlinearities traditionally leads to a power-independent $n_{2}$ response for dielectric materials, this description might fail for nontransparent graphene, in which nonlinear effects tend to be mediated by significant power-dependent carrier dynamics. This might explain that nonlinear effects in graphene might not increase under larger powers as for traditional dielectric media. Instead, the $\gamma_{hybrid}$ parameter of graphene-covered waveguides might provide an effective nonlinear response that is only valid for a limited range of powers, and the absolute value of this effective response would appear to decrease with increasing powers according to our experiments with respect to prior works conducted at lower powers. This hypothesis is supported by the recent theoretical description of the nonlinear response of graphene [21,22,31], in which a strong link between the carrier dynamics in graphene and its nonlinear response has been established.

## 5. Design Rules for Optimizing Hybrid Nonlinear Waveguides Locally Coated with 2D Materials

One last question arising from this investigation is a practical one: are there some optimum conditions to leverage the graphene nonlinearity when exploiting graphene patches, or more generally 2D material patches, integrated within a passive photonic circuit? Let us suppose that a patch of 2D material (either graphene or any other 2D material) of length $L_{hybrid}$ is locally positioned on top of a longer passive and low loss waveguide, which exhibits a positive (and relative low) nonlinear parameter value $\gamma_{bare}$, associated to the bare waveguide. Taking four-wave mixing as an example, two scenarios are considered, one for each sign associated with the nonlinear coefficient $\gamma_{hybrid}$ along the 2D material-covered section of the waveguide. In order to derive simpler analytical expressions (see Appendix A), we only consider a 2D material-covered section of waveguide followed by a bare section of waveguide without 2D material. Compared to the case used in our experiments, it is equivalent to neglect the contribution from the first short bare waveguide section, which just adds an offset to the nonlinear effect measured from the whole waveguide.

Our analysis (see Appendix A) shows that, under these conditions, the trend observed on the idler generation as a function of the 2D material length critically depends on whether the nonlinear parameter ratio, $|\gamma_{hybrid}|$/$\gamma_{bare}$, between the hybrid 2D material/dielectric waveguide and the bare waveguide is greater or lower than the product $L_{eff,bare}\times \alpha_{hybrid}$, with $L_{eff,bare}$ the effective length of the bare section of waveguide after the 2D material. Note that the quantity $L_{eff,bare}\times \alpha_{hybrid}$ eventually equals the $\alpha_{hybrid}$/$\alpha_{bare}$ loss ratio of the hybrid waveguide with respect to bare waveguide, for long passive circuits.

More quantitatively, we show that the 2D material patch increases the net idler generation of the overall waveguide as long as one of the following conditions are met, depending on the sign of $\gamma_{hybrid}$: (4)$$\left\{\begin{array}{c} \mathrm{Case}\gamma_{hybrid}>0:\;\frac{\gamma_{hybrid}}{\gamma_{bare}}>\frac{3}{2}\alpha_{hybrid}L_{eff,bare}\\ \mathrm{Case}\gamma_{hybrid}<0:\;\frac{|\gamma_{hybrid}|}{\gamma_{bare}}>3\alpha_{hybrid}L_{eff,bare} \end{array}\right..$$

In both cases, there is an optimum 2D material length that optimizes the nonlinear idler generation, which is found to be
(5)$$L_{hybrid}^{max}=\frac{1}{\alpha_{hybrid}}\log\left(3\left(1-\frac{\gamma_{bare}\alpha_{hybrid}L_{eff,bare}}{\gamma_{hybrid}}\right)\right).$$

Eventually, if the 2D material nonlinear response is strong enough to satisfy Equation (Equation 4), then the 2D material nonlinearity can benefit the net nonlinear effect accumulated across the entire waveguide. Otherwise, the net nonlinear effect decreases for increasing 2D material length, because the loss penalty is larger than the nonlinear enhancement locally imparted by the 2D material. Both scenarios are illustrated in Figure 6, for either sign of $\gamma_{hybrid}$.

We note that the constraint set by Equation (Equation 4), on $\gamma_{hybrid}$ is harder to meet when the latter has an opposite sign to that of the bare waveguide (Figure 6b), with respect to the situation where both the 2D material nonlinearity and that of the underlying waveguide material jointly contribute to the nonlinearity (Figure 6a). Consistently, the optimum 2D material length given by Equation (Equation 5) also tends to be larger when the sign of the nonlinear parameters are different. In the case in which $\gamma_{hybrid}<0$, we also observe the existence of a minimum idler power, which is almost canceled out, for a length of 2D material that compensates for the opposite nonlinear contribution from the bare waveguide.

In our experiments, $L_{eff,bare}\times \alpha_{hybrid}∼32$. Therefore, a net increase of idler intensity with graphene could appear with $\gamma_{hybrid}$ values greater than $\gamma_{hybrid}∼48$ m^−1^W^−1^ (if $\gamma_{hybrid}>0$) or than $\gamma_{hybrid}∼-96$ m^−1^W^−1^ (if $\gamma_{hybrid}<0$). Our experimental results, and the comparison to more refined simulations (which yield $\gamma_{hybrid}∼-10$ m^−1^W^−1^), indicate that the benefit of graphene, in our chip, is not only overcompensated by its loss penalty, but is also not enough to compensate for the nonlinear effect induced by the surrounding sections of Si${}_{3}$N${}_{4}$.

One simple solution to experimentally observe a benefit of graphene on photonic circuits would be to shorten the surrounding bare waveguides or, more practically, to decrease its nonlinearity. For instance, other waveguide platforms might be used, such as SiO${}_{x}$ glass waveguides with 10 times lower nonlinearity that might boost the ratio $\gamma_{hybrid}/\gamma_{bare}$ via a reduction of $\gamma_{bare}$ for the surrounding waveguide. Without changing the material platform, one could alternatively decrease the $\gamma_{bare}$ parameter of the surrounding waveguide by enlarging its cross-section while keeping it locally small to increase the graphene–light interaction only where graphene is to be integrated. Finally, other 2D materials with a better loss/nonlinearity trade-off (i.e., a higher $\gamma_{hybrid}$/$\alpha_{hybrid}$), like graphene oxide [32,33], might be better suited for nonlinear applications.

## 6. Conclusions

By transferring graphene onto locally unclad Si${}_{3}$N${}_{4}$ waveguides, we could control the position and length of submillimeter-scale graphene patches integrated onto passive and low-loss mature photonic waveguides so as to investigate its potential for integrated nonlinear photonics at telecom wavelengths. Degenerate four-wave mixing experiments were conducted in a pulsed regime by using pump and probe peak powers up to 7 W coupled, i.e., at least a factor of 10 times higher than in prior FWM experiments. By measuring the generated idler intensity as a function of the graphene length and pump power, and comparing the results with simulations, we could separate the impact of the loss increase and nonlinear enhancement locally imparted by graphene, which was acknowledged from the early days to be a significant trade-off for this 2D material [34]. From our analysis, the net reduction of the four-wave mixing efficiency induced by graphene can be attributed to two combined effects: (1) a negative nonlinear parameter of the hybrid graphene/Si${}_{3}$N${}_{4}$ section ($\gamma_{hybrid}∼-10$ m^−1^W^−1^) locally increased by an order of magnitude with respect to that of the underlying waveguide, and (2) a strong linear loss of graphene reducing the contribution of the following Si${}_{3}$N${}_{4}$ section without graphene. A comparison of our results with the literature showed that the 10 times nonlinear enhancement found in our case is relatively low, and cannot be fully explained by the lower interaction between the guided mode profile and graphene. These results suggest that the carrier-mediated effective nonlinear response of graphene strongly depends on the experimental conditions used, and effectively decreases upon larger powers, thereby limiting the absolute nonlinear effects afforded by this material under practical use. Our results thus help to clarify the conditions under which graphene could be used for nonlinear applications. Finally, we highlight a simple rule of thumb, relying on the nonlinearity/loss tradeoff to assess whether graphene and, more generally, 2D material patches might be able to benefit and locally enhance the nonlinear response of otherwise passive photonic circuits. Our work thus contributes to clarifying the potential of hybrid 2D material waveguides for nonlinear applications.

## Figures and Tables

**Figure 1 nanomaterials-13-00451-f001:**
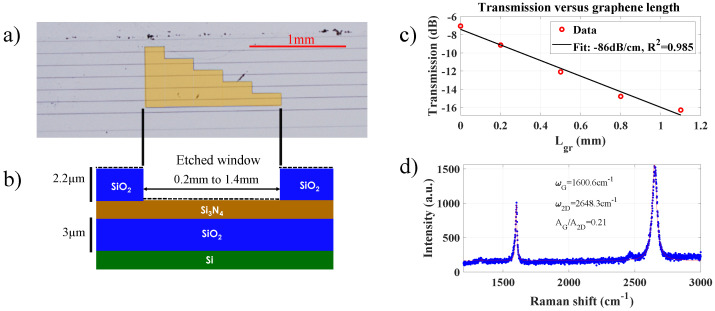
(**a**) Top-view optical microscope image of the waveguides with the window highlighted in orange, where graphene covers the waveguides. (**b**) Schematic view of the structure cross-section, around the etched cladding window. (**c**) Total transmission of the waveguides as a function of the graphene length covering the Si${}_{3}$N${}_{4}$ waveguide (i.e., the length of the associated cladding window). (**d**) Raman spectrum measured (blue dots) and fitted (red line) of the graphene in direct contact with the waveguide. The $A_{G}/A_{2D}$ is the ratio of the peak area obtained from the fit, and allows one to check that the graphene is monolayer [26].

**Figure 2 nanomaterials-13-00451-f002:**
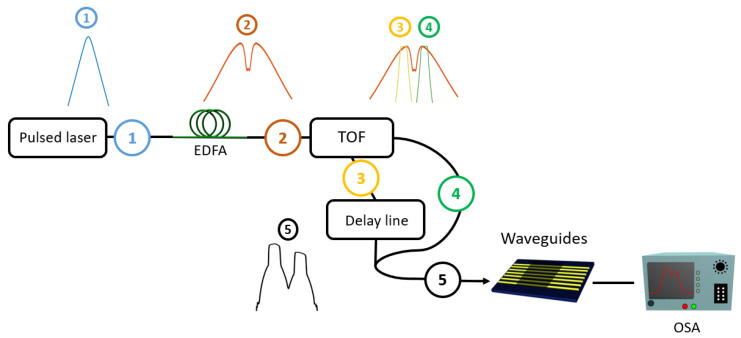
Experimental setup used to perform pulsed FWM in the graphene-covered waveguides. (1) The pulsed laser delivers Fourier-limited pulses ($\tau_{FWHM}=2ps$). (2) The pulses are amplified and spectrally broadened upon propagating in the erbium-doped fiber amplifier (EDFA). A tunable optical filter (TOF), WaveShaper 4000s^®^ is used to filter and split the signal into two channels: (3) the pump, $\lambda_{p}=1545\;nm$ with a width of Δλ=2 nm and (4) the probe, $\lambda_{s}=1549\;nm$ with a width of Δλ=2 nm. A delay line is added on the pump channel to synchronize the two signals reunited in (5) just before coupling to the waveguides. An optical spectrum analyser (OSA) then measures the spectrum after propagation in the graphene-covered waveguides.

**Figure 3 nanomaterials-13-00451-f003:**
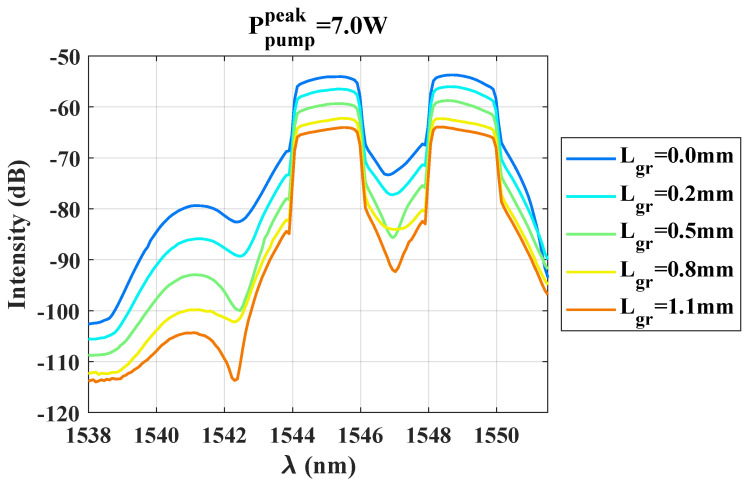
Measured spectrum for each graphene length interacting with the waveguide, and for a peak power $P_{pump}=P_{probe}=7\;W$. The idler (lobe around λ=1541 nm) is generated by the four-wave mixing process occurring along the waveguides.

**Figure 4 nanomaterials-13-00451-f004:**
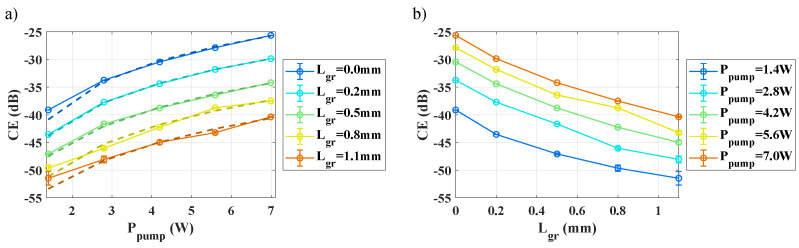
Measured *CE* (as per Equation (Equation 2), dB scale) as a function of (**a**) the coupled peak pump power, for different graphene lengths and with a quadratic fit (dotted line), and (**b**) the graphene length for different coupled peak pump powers. The coupled peak probe power is fixed at $P_{probe}=7\;W$. Error bars calculated from the spectrum noise are added, taking into account an error at (1σ).

**Figure 5 nanomaterials-13-00451-f005:**
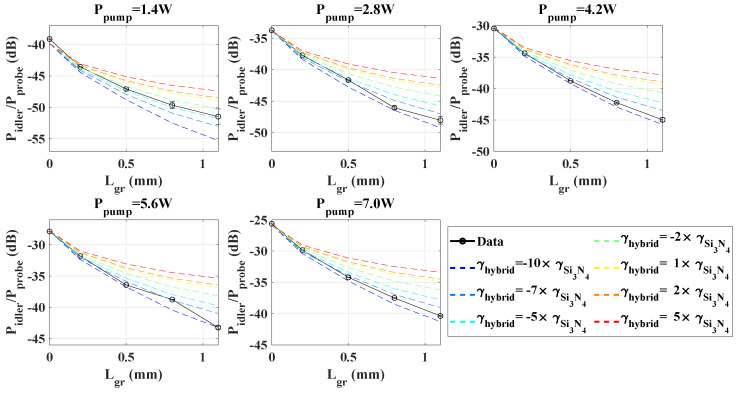
Measured *CE* (in black) as per Equation (Equation 2), versus the graphene length for different peak pump powers (in each subfigure). The dashed lines correspond to the simulation results, each considering a different nonlinear parameter for the hybrid graphene/Si${}_{3}$N${}_{4}$ section (expressed in units of $\gamma_{Si_{3}N_{4}}=1$ m^−1^W^−1^).

**Figure 6 nanomaterials-13-00451-f006:**
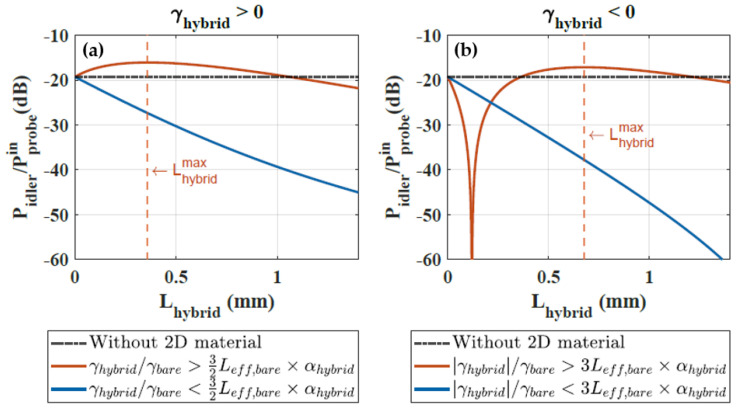
Idler power output divided by probe input power versus the 2D material length for (**a**) positive values of $\gamma_{hybrid}$ with $\gamma_{hybrid}/\gamma_{bare}=100$ (orange) and $\gamma_{hybrid}/\gamma_{bare}=5$ (blue), and (**b**) taking negative values of $\gamma_{hybrid}$ with $\gamma_{hybrid}/\gamma_{bare}=-120$ (orange) and $\gamma_{hybrid}/\gamma_{bare}=-1$ (blue). In both cases, $L_{eff,bare}\times \alpha_{hybrid}=32$ and the dotted black line at −20
dB corresponds to the idler generated without 2D material on the waveguide. The pump power is 7 W.

**Table 1 nanomaterials-13-00451-t001:** Table summarizing the parameters of the clad Si${}_{3}$N${}_{4}$ waveguide and the hybrid graphene/Si${}_{3}$N${}_{4}$, as used in the simulations.

Parameter	Clad Si${}_{3}$N${}_{4}$ Waveguide	Hybrid Graphene/Si${}_{3}$N${}_{4}$ Waveguide
Linear loss α	0.5 dB cm^−1^	86 dB cm^−1^
Dispersion $\beta_{2}$	−1.2 × 10^−25^ s^2^m^−1^	−2.1 × 10^−25^ s^2^m^−1^
Effective area	1.06 μ m	1.00 μ m
Nonlinear coefficient γ	1 W^−1^m^−1^	Determined below
Total insertion losses	6 dB	
Loss at the etched interface	0.5 dB per facet	

**Table 2 nanomaterials-13-00451-t002:** Table of different references reporting nonlinear experiments in hybrid graphene/ dielectric waveguides at telecom wavelength. For the nonlinear coefficient along the hybrid waveguide, $\gamma_{hybrid}$, the symbol * indicates a case in which the sign was not extracted. For the reference [20], the symbol ** is used to highlight that the nonlinearity of graphene is modelled considering carrier refraction. The comparison with $\gamma_{hybrid}$ is therefore not direct.

Ref.	Type	Waveguide	$A_{\mathrm{eff}}$	$L_{\mathrm{gr}}^{\max}$	$P_{\max}$	τ	$\alpha_{\mathrm{hybrid}}$	$\gamma_{\mathrm{hybrid}}$	$n_{2,\mathrm{graphene}}^{\mathrm{eff}}$	$|\gamma_{\mathrm{hybrid}}/\alpha_{\mathrm{hybrid}}|$
		Core	($\mu m^{2}$)	(mm)		(ps)	(dB cm${}^{-1}$)	$\left(m^{-1}W^{-1}\right)$	$\left(m^{2}W^{-1}\right)$	$\left(W^{-1}\right)$
[12] (2016)	SPM	Si	∼0.1 $\mu m^{2}$	0.2	1.7 W	1.2 ps	1320	−1700	$-10^{-13}$	0.0559
[10] (2017)	FWM	Si${}_{3}$N${}_{4}$	∼0.5 $\mu m^{2}$	0.1	10 mW	CW	400	4000 *	N/A	0.4343
[29] (2017)	SPM	Si	0.144 μm	0.2	1000 W	80 fs	520	1600	$+10^{-13}$	0.1336
[20] (2018)	SPM	Si${}_{3}$N${}_{4}$/SiO${}_{2}$	N/A	1.1	2.7 W	3 ps	200	N/A **	N/A	N/A
[16] (2019)	FWM	Si	0.144 μ m	0.2	140 mW	CW	670	∼1540	$+10^{-13}$	0.0998
[30] (2019)	SPM	Si	0.16 μ m	0.06	∼10 W	1.5 p s	200	510	N/A	0.1107
This work	FWM	Si${}_{3}$N${}_{4}$	1 μm	∼1	7 W	2 ps	86	∼−10	$-10^{-14}$	0.0050

## Data Availability

The authors confirm that the data supporting the findings of this study are available within the article and its Appendix A.

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
