# Peer review of "Pulsed Four-Wave Mixing at Telecom Wavelengths in Si3N4 Waveguides Locally Covered by Graphene"

_nanomaterials, 2023, doi:10.3390/nano13030451_

Round 1
Reviewer 1 Report
In this work the conversion efficiency (CE) of a four-wave mixing (FWM) experiment with pulsed pumping in silicon nitride waveguides covered by a graphene layer is studied experimentally. The experimental results are analyzed based on simulations leading to two major conclusions: (1) graphene features a negative nonlinearity; and (2) the nonlinearity enhancement due to graphene does not translate into an improvement of the CE mainly owing to the concomitant loss increase. These results fully agree with those reported in Ref. [20], where the pulse broadening efficiency due to self-phase modulation in graphene-cladded silicon-nitride waveguides was investigated. I have the following concerns:
1) Loss is measured under CW at low powers (see line 105), in contrast to the conditions where the FWM experiments are performed. Have the authors checked that losses remain unaltered at higher powers?
2) The authors model their FWM experiments perturbatively based on a third-order graphene's nonlinear response. At the same time, they acknowledge that such a perturbative approach might not be accurate given the role of photoexcited carriers in regard to graphene's nonlinearity (see lines 40-44 and 295-298). Could the authors compare the intensities employed in their experiments with graphene's saturation intensities? Is the perturbative description of these experiments justified in view of this comparison?
3) Equation (3) neglects the non-degenerate FWM process, i.e., ooP + ooS --> ooI1 + ooI2, where ooI1 would correspond to a wavelength 1541 nm and ooI2 to 1553 nm (not included in Fig. 3). Could the authors justify this assumption?
4) Figure 5 shows that the negative effective nonlinear coefficient of the hybrid waveguides might decrease (i.e., increase in absolute value) with pump power, as the authors mention in lines 301-302. They also suggest that carriers dynamics might explain this power dependence. However, saturation effects due to photoexcited carriers would reduce, in principle, the absolute value of the effective nonlinearity at higher powers. Could the authors comment on this point?
5) In regard to Table 2, I have two remarks:
a) gamma_hybrid corresponding to Ref. [20] could also be derived, e.g., by considering its Eq. (6) at low powers, i.e., in the perturbative regime.
b) If sign(gamma_hybrid) was not extracted in Ref. [16], how n_{2,graphene}>0 was derived?
6) The authors should indicate that the modest enhancement shown in Fig. 6 at the optimal L_hybrid with respect to the bare waveguide would be an upper boundary since gamma_hybrid and alpha_hybrid are not independent parameters in practice, as assumed in Fig. 6.
Other remarks
7) Since disperion terms of order n>2 seem to be neglected attending to deltaj = beta2 DELTAomega (see line 186), I would suggest replacing the sum in Eq. (3) by its term corresponding to n = 2.
8) In Table 1: Is the change in beta2 mainly due to the cladding removal or does graphene affect beta2 too?
9) Y-axis units in Fig. 5 should be dB.
10) Check Y-axis label in Fig. 6. Should it be P_idler/P_probe instead of P_idler?
11) In line 31: li(t)erature.
Although the results reported support the conclusions of this work, the above concerns 1-4) motivate a revision of the manuscript by the authors.
Reviewer 2 Report
The manuscript "Pulsed four-wave mixing at Telecom wavelengths in Si3N4 waveguides locally covered by graphene" by Demongodin et al. experimentally and theoretically studied the near degenerate four wave mixing in a graphene covered Si3N4 waveguide, and found the condition when the idler signal can benifit from graphene's optical nonlinearity. The topic is very interesting and is also important for a graphene integrated photonic circuits. The condition when the graphene's nonlinearity plays an important role is practically meaningful. The experimental design and the data analysis are reasonable. I recommand its publication in Nanomaterials.
Reviewer 3 Report
Publication: "Pulsed four-wave mixing at Telecom wavelengths in Si3N4 waveguides locally covered by graphene" is a very interesting presentation of research on Si3N4 waveguides. I have two comments about the work: The thickness of the SiO2 layer between SI and Si3N4 is not specified. Figure No. 5 is too small and therefore almost illegible (when using the printed version of the work) In my opinion, after meeting the above-mentioned comments, the work can be published.
Round 2
Reviewer 1 Report
Please, let me briefly comment some authors' responses, including a suggestion:
2. Even if the dependence of the absorption on power is not significant, nonperturbative effects (that cannot be described by means of an effective third-order nonlinear response) may still be observed, as shown in Ref.[20].
4. Thanks for clarifying this point. I had compared the values of the effective nonlinear coefficient at 1.4 W and, e.g., 4.2 W, which suggested a hypothetical enhancement of the effective nonlinearity at higher powers. The authors might consider incorporating their comment about the reliability of the measurement at 1.4 W included in the response letter into the main text to avoid this issue.
5. (a) Just a remark: At low powers, the modeling accounting for carrier photorefractive effects should also lead to Eq. (3), but with a nonlinear coefficient defined in terms of carrier parameters (see Ref. [21] for more information).
6. Just to clarify my concern: I did not intend to question the usefulness of this analysis. On the contrary, I agree with the authors that dealing with nonlinear problems separately, and including further constraints subsequently, is generally a quite convenient approach. I tried to point out that the constraint imposed by the link between graphene's loss and nonlinear coefficients, see the equation included in the authors' answer 5(a), should also eventually be considered.
The authors have successfully addressed my concerns and therefore I recommend now this manuscript for publication.
